# `DrJAX`: Scalable and Differentiable MapReduce Primitives in JAX

Keith Rush [* 1]   Zachary Charles [* 1]   Zachary Garrett [1]   Sean Augenstein [1]   Nicole Mitchell [1]

## Abstract

We present `DrJAX`, a JAX-based library designed to support large-scale distributed and parallel machine learning algorithms that use MapReduce-style operations. `DrJAX` leverages JAX's sharding mechanisms to enable native targeting of TPUs and state-of-the-art JAX runtimes, including Pathways (Barham et al., 2022). `DrJAX` embeds building blocks for MapReduce computations as primitives in JAX. This enables three key benefits. First, `DrJAX` computations can be translated directly to XLA HLO, enabling flexible integration with a wide array of ML training platforms. Second, `DrJAX` computations are fully differentiable. Last, `DrJAX` computations can be interpreted out to existing batch-processing compute systems, including traditional MapReduce systems like Apache Beam and cross-device compute systems like those powering federated learning applications. We show that `DrJAX` provides an easily programmable, performant, and scalable framework for parallelized algorithm development.

## 1. Introduction

The ability to scale abstractly written compute-intensive programs across large distributed compute environments is a key factor in the success of modern machine learning (ML). This is crucial for ML computations involving large language models, which are often too large to fit on a single compute node. Another key facet of modern ML software is the general ease with which computations can be written and optimized. Techniques such as automatic differentiation (AD) and just-in-time (JIT) compilation have enabled frameworks such as PyTorch (Paszke et al., 2019), TensorFlow (Abadi et al., 2016), and JAX (Bradbury et al., 2018) to scale to larger and more complex ML workloads.

---
[*]Equal contribution [1]Google Research. Correspondence to: Keith Rush <krush@google.com>, Zachary Charles <zachcharles@google.com>.

Accepted to the Workshop on Advancing Neural Network Training at International Conference on Machine Learning (WANT@ICML 2024).

These software frameworks generally focus on enabling parallelism for their most common use case: computation of a function's derivative across a batch of inputs. This computation is typically parallelized in two well-defined manners: across the batch dimension (i.e. data parallelism), and within the computation of the derivative at a single example (i.e. model parallelism). Data parallelism is extremely common across ML frameworks. Model parallelism is more complex, but has seen an explosion of progress in recent years (Gholami et al., 2018; Shazeer et al., 2018; Jia et al., 2019; Lepikhin et al., 2020; Xu et al., 2021) which has enabled the training and deployment of significantly larger models than previously possible (e.g. foundation models).

However, many sub-fields of ML do not fit neatly into this description, and instead employ parallelism over higher-level partitioned structures of data: in meta-learning (where data is partitioned across tasks) (Finn et al., 2017); in group-level differential privacy (where data is partitioned over discrete groups whose individual contributions to algorithm outputs are information-theoretically bounded) (Dwork, 2010); in model merging (Li et al., 2022) or "model soup" algorithms (where data, in the form of hyperparameters, is partitioned across model copies) (Li et al., 2022; Wortsman et al., 2022); in federated learning (where data is partitioned across clients who avoid directly sharing data) (McMahan et al., 2017); and in optimization with intermittent communication (where data is partitioned across model replicas) (Mangasarian and Solodov, 1993; Zinkevich et al., 2010; Zhang et al., 2016). The algorithms studied in these nominally different areas share many commonalities. An especially common factor is the use of the MapReduce (Dean and Ghemawat, 2004) programming paradigm, mapping parallel model training steps over partitioned data before invoking a reduction function. Data parallelism is a special case: we simply map and reduce over batches of data. In other applications, MapReduce is applied to coarser groups of data (e.g. multiple batches), which may additionally encode semantic structure of the underlying data.

While ML frameworks provide scalability, flexibility, and efficiency for data- and model-parallelism, an algorithm author who wishes to program over partitioned data in a parallel manner finds themselves in an awkward position. For example, frameworks for federated learning (e.g. Ingerman and Ostrowski (2019); Ziller et al. (2021); Ro et al. (2021);

He et al. (2020)) offer parallelism over clients, but either do not support other ML use cases discussed above, or do not focus on large-scale datacenter performance. Underlying ML frameworks often offer powerful parallelism primitives (e.g. `jax.shard_map`), but these often assume (as is the case for `jax.shard_map`) that the specification of all model shardings and physical resources is known to the code author (Shazeer et al., 2018). By contrast, in `DrJAX` we wish to abstract out MapReduce-style computations, allowing them to be defined in terms of model forwards and backwards passes (for example) that are already sharded. Finally, we note that bridging research and production often requires translating computations (e.g. from JAX) to production platforms.

An ideal authoring surface for ML algorithms using MapReduce operations provides several features simultaneously: performant and scalable datacenter performance; the ability to decouple logical partitioning of data (number of groups of data to parallelize over) from physical compute (number of compute nodes); easy and extensible algorithm expression; JIT compilation and AD; and the capacity to translate algorithms to alternative infrastructure.

**Contributions.** We present `DrJAX` (**D**ifferentiable Map**R**educe **JAX**) a software library that brings the benefits of modern large-scale machine learning software – sharding, easy-to-use JIT compilation, and AD – to MapReduce-style algorithms operating on partitioned data. `DrJAX` embeds a simple mapping and reduction programming model, by decomposing computations into *differentiable* building blocks (Section 2). We fully implement this programming model in JAX by embedding these building blocks via JAX's `Primitive` mechanism (Section 3). This allows `DrJAX` to use powerful features like sharding and JIT compilation to XLA. For example, `DrJAX` can shard computations over data partitions, model, and within-data partitions simultaneously across physical and logical meshes of devices. Because `DrJAX` is essentially a careful programming of parallel algorithms in XLA, `DrJAX` can leverage advances in distributed datacenter training like GSPMD (Xu et al., 2021) and Pathways (Barham et al., 2022). We showcase the scaling and runtime benefits of `DrJAX` across a suite of large language model training experiments (Section 4).

JAX's `Primitive` mechanism also enables forward- and reverse-mode differentiation which `DrJAX` leverages to provide full differentiability of its MapReduce-style programs. By implementing the AD framework of Rush et al. (2023), we ensure that the derivative of a `DrJAX` program is simply another `DrJAX` program. This allows `DrJAX` and its AD system to be interpreted out to other platforms for parallel machine learning, including systems with strong guarantees about data locality and privacy (Section 5).

**Related work.** `DrJAX` draws inspiration from three main areas of software. First, `DrJAX` directly utilizes the programming model of MapReduce (Dean and Ghemawat, 2004) and its evolution in Apache Beam (Foundation), highlighting the power of focusing on replications, mappings and reductions on parallelized collections of data.

Second, much of `DrJAX`'s treatment of machine learning and automatic differentiation is influenced by modern ML frameworks such as PyTorch (Paszke et al., 2019) and TensorFlow (Abadi et al., 2016). `DrJAX`'s design is directly based on the functional-first nature of JAX (Bradbury et al., 2018). Several libraries implemented in JAX leverage similar (though not identical) mechanisms for ensuring scalability, notably the Praxis library for neural network layers (Google).

Finally, `DrJAX` is inspired by frameworks for federated learning. Without intending to be exhaustive, examples of such frameworks include: PySyft (Ziller et al., 2021), FedJAX (Ro et al., 2021), FedScale (Lai et al., 2022), FedML (He et al., 2020), Flower (Beutel et al., 2020), FLUTE (Dimitriadis et al., 2022), FL_Pytorch (Burlachenko et al., 2021), and FATE (Liu et al., 2021). In particular, `DrJAX`'s programming model was significantly influenced by TensorFlow Federated (Ingerman and Ostrowski, 2019) and its *federated computations* (Charles et al., 2022). Moreover, `DrJAX` uses the AD framework for federated computations proposed by Rush et al. (2023) to extend AD to MapReduce computations more broadly.

## 2. System Design

`DrJAX` is designed with two key ideas in mind. First, the types of parallel computing for machine learning discussed above can all be viewed as applications of MapReduce-style computations that use functions like model forward and backwards passes as black-box subroutines. Second, we can differentiate through MapReduce computations by using the AD techniques proposed by Rush et al. (2023). While their framework, federated AD, was motivated by federated learning, it can be directly adapted into a mechanism to apply AD to MapReduce computations. By combining standard AD techniques (e.g. computation graphs (Bauer, 1974)) with proper accounting of how data moves between logical partitions, we can express the derivative of a MapReduce computation as another MapReduce computation.

`DrJAX` operates on *partitioned* and *non-partitioned* values. A partition represents the data that we would like to perform MapReduce-style computations over. A non-partitioned value conceptually represents a singleton. We denote a non-partitioned value by $v$, and a partitioned value as $\mathbf{v} = [v_1, \ldots, v_n]$, where all $v_i$ are elements of the same space. Note that the value of $n$ depends on the partition, but the

ordering within the partition is arbitrary.

The inputs and outputs of `DrJAX` computations can include non-partitioned and partitioned values. Unlike classical MapReduce treatments, we do not assume that a computation will use a reducer. We consider a specific class of computations that can be built from the following computations, which we refer to as **building blocks**.

1. `broadcast`: Creates a partitioned value in which all groups have the same value (ie. $\text{broadcast}(v) = [v, v, \ldots, v]$).

2. `map_fn`: Applies a function $f$ across a partition (ie. $\text{map\_fn}(f, \mathbf{v}) = [f(v_1), f(v_2), \ldots, f(v_n)]$).

3. `reduce_sum`: Sums over a partitioned value, returning a non-partitioned value (ie. $\text{reduce\_sum}(\mathbf{v}) = \sum_{i=1}^{n} v_i$).

This class is sufficiently expressive to include many parallel algorithms of interest, including parallel model-agnostic meta learning (MAML) (Finn et al., 2017), parallel and local SGD (Mangasarian and Solodov, 1993; Zinkevich et al., 2010), federated averaging (FedAvg) (McMahan et al., 2017), Branch-Train-Merge (Li et al., 2022), DiLoCo (Douillard et al., 2023), and many others.

For our purposes, the key fact about these computations is that they are *closed* under MapReduce AD. In particular, let $f : x \mapsto y$ where $x$ is any collection of partitioned and non-partitioned inputs, and $y$ is non-partitioned. If $f$ can be composed from the building blocks above, then Rush et al. (2023) show that the function $\nabla f : x \mapsto dy/dx$ can also be expressed in terms of the building blocks above. Moreover, this can be done in a programmatic fashion. We will refer to this as *MapReduce AD* in the sequel.

This leads to our key observation: If we embed the building blocks above into JAX in a suitable manner, then we can (1) lower these computations to data structures accepted by performant data center runtimes, (2) implement MapReduce AD by appropriately delegating to JAX's AD, and (3) preserve partition information to enable translation to other production ML systems. `DrJAX` does just this, embedding the building blocks into JAX in a JIT-compatible manner. `DrJAX` also provides implementations of Jacobian-vector and vector-Jacobian products of the building blocks. This allows `DrJAX` to perform forward- and reverse-mode AD on MapReduce computations by delegating to JAX's forward- and reverse-mode AD.

**Authoring surface.** `DrJAX` code is almost entirely JAX code, with two general exceptions. First, there are the building blocks above. Second, `DrJAX` code must specify how many groups are in a partition during the invocation of the computation. To see this, consider the code in Snippet 1, which simply broadcasts a value across a partition, doubles the value in each group, and takes a sum over the partition.

```python
import drjax

def broadcast_double_and_sum(x):
  y = drjax.broadcast(x)
  z = drjax.map_fn(lambda a: 2*a, y)
  return drjax.reduce_sum(z)
```

*Snippet 1.* An incomplete `DrJAX` program, which broadcasts $x$ to a partition, doubles the value held by each group, and then sums over the partition. The program must know the partition size to correctly compute the desired result.

To compute the result, `DrJAX` needs to know the size of the partition. The user has to give this information to the `DrJAX` programs. For example, Snippet 2 modifies Snippet 1 to include explicit information about the partition size (ie. the number of groups in the partition).

```python
@drjax.program(partition_size=3)
def broadcast_double_and_sum(x):
  y = drjax.broadcast(x)
  z = drjax.map_fn(lambda a: 2*a, y)
  return drjax.reduce_sum(z)
```

*Snippet 2.* A basic `DrJAX` program, with a decorator specifying the partition size. With this information, `DrJAX` can determine that the program should return $6x$.

## 3. Implementation

We now discuss `DrJAX`'s implementation in JAX, in particular how it represents partitioned values and implements computations on them. We also discuss how we ensure `DrJAX` computations are effectively sharded across data center runtimes, and how `DrJAX` can implement MapReduce AD. While we focus on the programming model above, we note `DrJAX`'s lower-level implementation can be used for much more general distributed and even hierarchical processing and sharding patterns.

**Partioned values.** `DrJAX` represents both partitioned values as arrays with an extra leading dimension indicating the number of groups associated to them. Compared to partitioned values, non-partitioned values have an extra leading axis of cardinality equal to the number of groups. Given an $(d + 1)$-dimensional array $x$, the $i$-th component $x[i, \ldots]$ is the $d$-dimensional array held by the $i$-th group. Figure 1 gives an example of this representation.

All JAX values are essentially represented as structures whose leaf nodes are arrays (referred to as pytrees in JAX), which `DrJAX` carries forward. A partitioned structure is a

structure (ie. a pytree) of partitioned arrays. For example, Figure 2 gives an example of a partitioned structure with multiple leaf arrays.

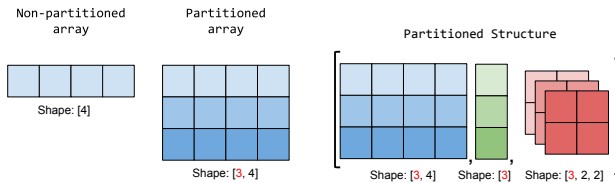

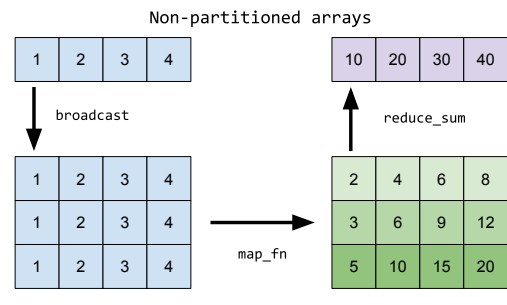

*Figure 1.* DrJAX's representation of a non-partitioned array (left) and an array partitioned over 3 groups (right).

*Figure 2.* A partitioned structure in DrJAX with 3 groups. Each leaf is a partitioned array.

*Figure 3.* A high-level depiction of DrJAX building blocks operating on and transforming non-partitioned and partitioned arrays.

**DrJAX computations.**    Since partitioned values are represented as JAX arrays, DrJAX computations must operate on JAX arrays. Other goals of DrJAX, like scalability, data center performance, and enabling differentiability, inform how DrJAX operates on arrays. We address these simultaneously by leveraging JAX's Primitive mechanism.

Briefly, DrJAX defines the building blocks above at decorator-installation time. These building blocks are processed *symbolically* by functional transformations in JAX. DrJAX registers the behavior of these operators under the action of these transformations, providing JAX with the necessary information to (1) lower DrJAX-defined functions wholesale to XLA HLO, (2) shard intermediate tensors in a maximally efficient manner, and (3) transform JAX functions containing DrJAX code under operations including JIT compilation and differentiation.

Given the representation of partitioned values above, we can implement the building blocks via straightforward array operations:

1. broadcast: Tile an array over its leading axis.

2. map_fn: Apply a function pointwise over an array's leading axis.

3. reduce_sum: Sum an array over its leading axis.

We extend these to partitioned structures by applying them leaf-wise. DrJAX registers these implementations with JAX lowering logic. This ensures that DrJAX code is entirely replaced by JAX code by the time JAX dispatches logic to an XLA runtime. Other building blocks can be added to DrJAX by registering primitives in a similar fashion, or by defining them in terms of the building blocks above. For example, DrJAX provides a reduce_mean symbol which takes an average across groups in a partitioned array, which lowers to two calls to reduce_sum.

**Sharding DrJAX computations.**    By registering the primitives above, we ensure that compilers like GSPMD (Xu et al., 2021) can shard DrJAX computations across worker nodes. Critically, and distinct from paradigms such as jax.pmap, DrJAX decouples partition size from sharding computations. A partition of size $n$ is purely logical, and can be sharded across any number of $m$ workers as long as $m|n$. We want to ensure that, no matter how many workers we shard over, DrJAX computations are as efficient as possible. To do so, we only need to focus on how the building blocks above are sharded by compilers. Once this is done, we are free to compose with model- and data-parallelism provided by various JAX libraries.

While some DrJAX building blocks are trivially parallelizable (e.g. map_fn), compilers may not be able to detect this and generate efficient code. As noted by Xu et al. (2021) and Lepikhin et al. (2020), internal sharding annotations can dramatically affect the performance of a compiler targeting distributed execution. DrJAX uses static and dynamic sharding constraints to ensure that after compilation, the resulting computation will run efficiently in the data center. As we will see in Section 4, without these annotations, compilers like GSPMD do not optimally shard DrJAX computations, especially as the partition size increases.

**Derivatives of DrJAX computations.**    The last benefit of embedding building blocks as JAX primitives is that it gives us a straightforward way to take derivatives of DrJAX computations using AD. We refer to this as *MapReduce AD*. To do so, we only need to define the action of vector-Jacobian products (VJPs) and Jacobian-vector products (JVPs) on the DrJAX primitives. Rush et al. (2023) discuss how to compute these products, and show that their computation does not require any new building blocks. That is, the JVPs and VJPs of these primitives can be expressed in terms of the same set of primitives. With the JVPs and VJPs, we can now entirely rely on JAX's AD to do forward- and reverse-mode AD on computations involving these primitives. For

more details, see Section 5.

## 4. Scalability and Efficiency

We now present numerical evidence of the scalability and efficiency of `DrJAX`, by testing `DrJAX` in a distributed training setting. We perform multiple rounds of local SGD (Zhang et al., 2016) on transformer language models with 350 million (350M), 1 billion (1B), and 8 billion (8B) parameters. We use a causal language modeling loss and a sequence length of 512. In every round, we parallelize 4 local SGD steps, each with batch size 8, across some number of data groups from a partition before synchronization. We use a partitioned version of CCNews dataset, where news articles are partitioned according to their base URL domain. We use Dataset Grouper (Charles et al., 2023) to iterate over groups of data efficiently. In each round, we sample some number of data groups, which form a partition of some size. We vary the partition size proportionally to the number of workers used in the computation. To describe the scale of the experiments, Table 1 contains the maximum number of tokens processed and model parameters updated per round for each model.

For all experiments, we shard the training computation over some number of TPUv2s. The total number of TPU chips, $m$, is proportional to the partition size, $n$. For 350M, 1B, and 8B models we use $n$, $4n$, and $8n$ chips in total, respectively. This means that if we double the partition size, we also double the number of TPU chips used. We fully shard the computations across the workers, and additionally do model parallelism for the 1B and 8B models. For all experiments, our `DrJAX` computations are first compiled using GSPMD (Xu et al., 2021) and then delegated to the Pathways runtime (Barham et al., 2022).

**Weak scaling.** The *weak scaling* of a system refers to how its compute time changes as the workload and compute resources scale simultaneously. Generally, modern ML systems attempt to obtain near-constant weak scaling performance.[1] For `DrJAX`, we fix the model size and number of local SGD steps computed per data group in the partition, and vary the partition size and number of workers proportionally in order to vary the workload size. As discussed above, we scale the number of TPU chips used in our simulations linearly with respect to the partition size.

Figure 4 shows how training time of `DrJAX`-based local SGD scales as the partition size and number of TPU chips increase, across a range of model sizes. `DrJAX`-exhibits near-constant runtime for a fixed model size, even up to a pool of 128 or 512 workers. This is highly non-trivial.

[1]Constant performance is generally impossible due to overhead such as synchronization costs.

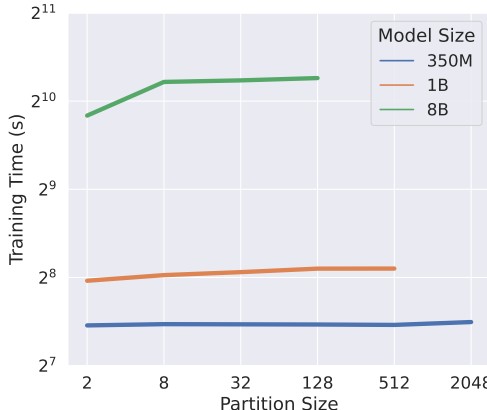

*Figure 4.* Total training time for 100 rounds of local SGD on various transformer language models sizes, with varying partition sizes.

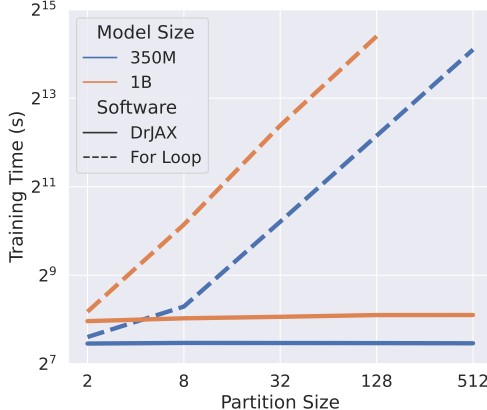

*Figure 5.* Total training time for 100 rounds of local SGD, with varying partition sizes. We implement local SGD using `DrJAX` and a python for-loop which we JIT compile.

Because local SGD involves parallel model training across workers, and for multiple steps per worker, the per-round workload size (in terms of total floating point operations) is at least as large as $4\times$(model size)$\times$(partition size). To see this, note that in each round, for each group in the data partition, we update a local model copy 4 times. As shown in Table 1, the largest workload for each model size involves updating over 1 trillion model parameters per round.

**JIT compilation alone is not enough.** ML research often involves writing custom training loops. A naive implementation of local SGD, often used for research in distributed training, is simply a double for loop that iterates over workers in the pool, and over the batches held by each worker. The outer loop here has no data dependency, meaning that the values returned by iterations of this loop are not processed as inputs to the next iteration. One might therefore imagine

*Table 1.* Maximum partition size, partition size, number of workers, number of tokens processed, and total floating point operations (FLOPs) when training with local SGD, for each model size. For simplicity, we only present FLOPs associated with the forward pass, using the approximation that a forward pass on a model of size $d$ uses $d$ FLOPs.

| Model Size | Partition Size | Num Workers | Tokens per Round | FLOPs per Round |
|---|---|---|---|---|
| 350M | 2048 | 2048 | $3.355 \times 10^7$ | $2.293 \times 10^{13}$ |
| 1B | 512 | 512 | $8.389 \times 10^6$ | $1.638 \times 10^{13}$ |
| 8B | 128 | 128 | $2.097 \times 10^6$ | $3.277 \times 10^{13}$ |

that a sufficiently advanced compiler could detect this fact, and parallelize worker training when possible (e.g. within resource constraints in the data center environment).

This can be a difficult task for a compiler. To illustrate this difficulty, we implemented a double for loop in place of `DrJAX`-based training (looping over workers, and over each worker's data). For both programs, we JIT-compiled the program, and provided identical input and output shardings to GSPMD and the XLA compiler stack. Though this stack is quite advanced and used to train many of the largest contemporary ML models, it does not recover the performance of `DrJAX` from this for-loop implementation. Indeed, round runtime scales linearly with the partition size (and therefore, the number of workers), as expected, rather than remaining constant, indicating an inability to use the increased resource scale allocated to the experiment.

**GSPMD alone is not enough.** A better way to parallelize across workers than the for-loop approach above is to implement `DrJAX`'s MapReduce building blocks and use a compiler like GSPMD (Xu et al., 2021) to do automated sharding of the program. This leads to the question: Do we need `DrJAX`'s internal sharding annotations to obtain weak scaling behavior, or can GSPMD alone fully and efficiently parallelize `DrJAX` computations? Given the relatively simple nature of local SGD's parallel processing patterns (heavily parallelized model training with infrequent synchronization), one might expect that isolating MapReduce building blocks as primitives with specially-designed sharding annotations is unnecessarily complex.

To test this, we took a `DrJAX`-based implementation of local SGD and removed all of `DrJAX`'s internal sharding annotations at function-tracing time, denoting this `DrJAX`-NS (`DrJAX` with no sharding). We then re-ran the simulations in Figure 4. The results in Figure 6 show that at present, these explicit sharding annotations play a crucial role in ensuring `DrJAX`'s weak-scaling behavior. `DrJAX`-NS computation times increased sublinearly but significantly faster than `DrJAX` computation times. Moreover, `DrJAX`-NS exhibited memory footprint scaling issues. We found that for sufficiently large model or worker pool sizes, `DrJAX`-NS eventually ran out of high-bandwidth memory. In particular, this occurred for the 1B model with 512 workers and for the

8B model with all tested numbers of workers; that is, at 2 workers and beyond.

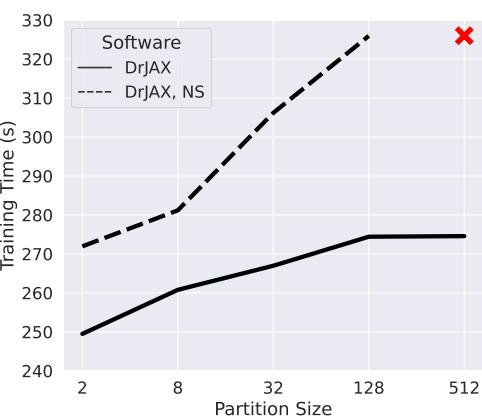

*Figure 6.* Total training time for 100 rounds of local SGD for the 1B model, with varying partition sizes. We implement local SGD using `DrJAX` with and without (`DrJAX`-NS) `DrJAX`'s sharding annotations. The red X represents the point at which the `DrJAX`-NS could not be sharded without triggering out of memory errors.

## 5. Interpreting **DrJAX** to Other Platforms

While data center performance is the primary goal of `DrJAX`, we wish to preserve the optionality to translate `DrJAX` computations into artifacts interpretable by other systems, such as federated learning systems (Bonawitz et al., 2019; Paulik et al., 2021; Huba et al., 2022). For example, if `reduce_sum` is only captured as a `jnp.sum`, then it may be difficult to tell whether this sum is intended to be *within* a partition group, or across groups in a partition. Below, we discuss how `DrJAX`'s implementation enables computing program representations that can be automatically translated to other platforms.

**Preserving partition information.** Recall from above that we implement MapReduce building blocks as JAX primitives, and build `DrJAX` computations out of these primitives. This has a key benefit when interpreting out to other systems: the ability to preserve information about how `DrJAX` building blocks are applied to partitioned data, which can inform things like cross-node communication

and computation boundaries within a production system.

JAX's `Primitive` mechanism allows users to inject new symbols into JAX itself, defining how these symbols behave under JAX's functional transformations like `jax.vmap` and `jax.grad`. These primitives are preserved in JAX's intermediate data structure, the `jaxpr`, which is usually later lowered into XLA HLO. By using a custom interpreter, we can instead generate and consume `jaxprs`. This custom interpreter can use special behavior when it encounters the `DrJAX`-defined symbols injected via the `Primitive` mechanism. This preserves information about data partitioning and operations within and across partitions, allowing us to translate `jaxprs` into computations that can be run by other platforms.

An example `jaxpr` is illustrative. In Snippet 3, we define a `DrJAX` program for computing the MAML loss from Finn et al. (2017). This is the loss of a model on some data *after* some number of steps of (stochastic) gradient descent. For our purposes, we do a single gradient descent step before evaluating the loss, using some user-specified learning rate. In the nomenclature of meta-learning, the data is partitioned across some number of *tasks*, and we assume we have access to some loss function $\texttt{loss}(x, y)$ where $x$ is the model, and $y$ is the task.

```
def maml_loss(model, lr, task):
  g = jax.grad(loss)(model, task)
  model = model - lr * g
  return loss(model, task)
```

*Snippet 3.* MAML loss

This loss is easily parallelized across workers using `DrJAX`, as in Snippet 4. The algorithm is straightforward: The model and learning rate are broadcast to every task, the MAML loss is computed using the model, learning rate, and task, and the resulting loss values are averaged.

```
@drjax.program(partition_size=3)
def parallel_maml_loss(model, lr, tasks):
  model = drjax.broadcast(model)
  lr = drjax.broadcast(lr)
  losses = drjax.map_fn(
    maml_loss, (model, lr, tasks))
  return drjax.reduce_mean(client_losses)
```

*Snippet 4.* Computing the average meta-learning loss over a data partition of size 3 via `DrJAX`.

To obtain a `jaxpr` representing this processing pattern, we provide the concrete shape and type of arguments. For brevity, we assume the model and tasks are scalars, and the loss function is the square loss (ie. $\texttt{loss}(x, y) = (x - y)^2$).

Given this information, JAX can generate a `jaxpr` representing Snippet 4. The result is in Snippet 5. The key takeaway is that this `jaxpr` preserves the `DrJAX`-defined primitives representing cross-machine communication, **broadcast** and **reduce_mean**, both of which are primitives registered by `DrJAX` in JAX. We can trace through the arguments in the `jaxpr` to see that the computation operates by (1) broadcasting values (the model and learning rate) across the partition, (2) calculating `loss` using the broadcast values, (3) taking a mean over the partition.

```
{ lambda ; a:f32[] b:f32[] c:f32[3]. let
    d:f32[1] = broadcast a
    e:f32[1] = broadcast b
    f:f32[1] = sub d e
    _:f32[1] = integer_pow[y=2] f
    g:f32[1] = integer_pow[y=1] f
    h:f32[1] = mul 2.0 g
    i:f32[1] = mul 1.0 h
    j:f32[1] = mul c i
    k:f32[1] = sub d j
    l:f32[1] = sub k e
    m:f32[1] = integer_pow[y=2] l
    n:f32[] = reduce_mean m
  in (n,) }
```

*Snippet 5.* `jaxpr` generated for `parallel_maml_loss`.

Interpreting the `jaxpr` to a production systems, especially distributed systems, such as TensorFlow Federated (Ingerman and Ostrowski, 2019) is now straightforward: all cross-machine communication is explicit, and the processing in-between communication is entirely local and can be extracted into standalone functions executed locally by compute nodes in the system.

**Integrating MapReduce AD.** As discussed in Section 2, the `Primitive` mechanism allows `DrJAX` to to specify the behavior of building blocks under JAX's functional transformations, including computing forward- and reverse-mode Jacobians (`jax.jacfwd` and `jax.jacrev`). This allows `DrJAX` to apply AD to MapReduce computations via the forward- and reverse-mode algorithms presented in Rush et al. (2023). For example, forward- and reverse-mode Jacobians of `broadcast` can be computed via `broadcast` and `reduce_sum`, respectively. `DrJAX` can therefore implement MapReduce AD without additional primitives. This means that the `jaxpr` of `DrJAX` computations that use MapReduce AD will contain JAX's standard AD symbols, along with `DrJAX`'s primitive set, ensuring that computations using MapReduce AD are still interpretable to other systems.

For example, Snippet 6 gives the `jaxpr` of the reverse-mode gradient of `parallel_maml_loss`) (ie. the deriva-

tive of the parallel MAML loss computation in Snippet 4). Again, we see that information about communication in the system is preserved. The jaxpr contains the primitives **broadcast**, **reduce_mean**, and **reduce_sum**, and which just as above, can be used by a custom interpreter to translate the jaxpr into a production system.

```
{ lambda ; a:f32[] b:f32[] c:f32[3]. let
    d:f32[1] = broadcast a
    e:f32[1] = broadcast b
    f:f32[1] = sub d e
    _:f32[1] = integer_pow[y=2] f
    g:f32[1] = integer_pow[y=1] f
    _:f32[1] = mul 2.0 g
    h:f32[1] = integer_pow[y=1] f
    i:f32[1] = integer_pow[y=0] f
    j:f32[1] = mul 1.0 i
    k:f32[1] = mul 2.0 h
    l:f32[1] = mul 1.0 k
    m:f32[1] = mul c l
    n:f32[1] = sub d m
    o:f32[1] = sub n e
    p:f32[1] = integer_pow[y=2] o
    q:f32[1] = integer_pow[y=1] o
    r:f32[1] = mul 2.0 q
    _:f32[] = reduce_mean p
    s:f32[1] = broadcast 1.0
    t:f32[1] = div s 1.0
    u:f32[1] = mul t r
    v:f32[1] = neg u
    w:f32[1] = mul c v
    x:f32[1] = mul 1.0 w
    y:f32[1] = mul 2.0 x
    z:f32[1] = mul y j
    ba:f32[1] = add_any u z
    bb:f32[] = reduce_sum ba
  in (bb,) }
```

*Snippet 6.* jaxpr generated for jax.grad(parallel_maml_loss).

## 6. Discussion

**Why MapReduce AD?** While features like scalability and efficiency are self-explanatory, the reader may be interested in *why* we wish to implement MapReduce AD, especially given the care required to interpret to production systems. In short, MapReduce AD makes expressing efficient algorithms easier (Rush et al., 2023). By way of analogy, AD has made the development of sophisticated neural network architectures significantly easier. Libraries can define the conceptually simpler forward-pass, and rely on AD to perform backpropagation. The result is often faster and less error-prone than hand-implemented gradient computations (Baydin et al., 2018).

Algorithms that operate on partitioned data can see similar benefits. For example, Snippet 4 contains a

DrJAX program used to compute the average MAML loss over tasks, as in meta-learning. By simply calling jax.grad(parallel_maml_loss), we immediately get a DrJAX program that computes the average MAML gradient over tasks. Using this, we can easily write an algorithm that efficiently parallelizes the MAML algorithm. Snippet 7 depicts this, defining implementing a parallel MAML algorithm by simply pairing jax.grad with an SGD update step.

```
@drjax.program(partition_size=3)
def parallel_maml_step(model, lr, tasks):
  g = jax.grad(parallel_maml_loss)(model,
    lr, tasks)
  return model - lr * g
```

*Snippet 7.* Implementing Parallel MAML via MapReduce AD.

**Self-tuning algorithms.** Another potential use case for MapReduce AD is creating self-tuning algorithms, which use AD to optimize algorithmic hyperparameters (e.g. *hypergradient descent*). By using MapReduce AD, we can automatically adjust hyperparameters that govern underlying optimization algorithms, but also adjust the MapReduce operations themselves. For example, Rush et al. (2023) show that the weights governing a weighted mean-based reduction can be learned in tandem with performing federated learning. Similarly, Wang et al. (2023) derive formulas for hypergradient descent on federated learning algorithms, but this process can be slow, error-prone, and algorithm-specific. By contrast, Rush et al. (2023) use an AD system (that inspired MapReduce AD, as we discuss in Section 3)) to do the same, without needing to derive algorithm-specific rules. More generally, MapReduce AD opens the door to a wide variety of self-tuning distributed and parallel algorithms.

**Conclusion.** By pairing differentiable MapReduce primitives with an easy-to-use front-end via JAX, performant building block implementations, and useful sharding information, we hope to accelerate research on distributed and parallel ML. Future work includes (1) generalizations of DrJAX to non-linear reductions, (2) extensions of DrJAX to more general types of data, including hierarchically partitioned data, and (3) mature DrJAX interpreters for specific production systems.

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
