# OpenReview forum: "DrJAX: Scalable and Differentiable MapReduce Primitives in JAX"
_ICML.cc/2024/Workshop/WANT — WANT@ICML 2024 Poster_

### Official Review · Reviewer_2iKU · 2024-06-12
**Specialised Primitives for Distributed Training in JAX**

**Confidence:** 5

**Summary:**

DrJAX is a library that adds a new way of distributing models and data over many workers, for instead in a datacentre setting. It does this by introducing a few new operations in the form of broadcast, map and reduce, which are implemented as JAX primitives that internally handle sharding and communication of data among workers. It also introduces a concept of partitioned values, where data known to be independent is made explicit. This would allow DrJAX to efficiently divide it among available compute resources. DrJAX in the paper is demonstrated using code examples and compared in experiments to approaches which rely on solely on compiler optimisations.

**Strengths:**

* The library has a clean integration with JAX and the associated XLA compiler stack, since it is implemented using custom JAX primitives.
* DrJAX appears to have a very simple intuitive API matching existing paradigms using map and reduce, allowing the user to scale a JAX program with a small amount of code modification.
* Significantly simpler than alternatives such as using JAX's sharding API.
* The task of sharding data and models in a distributed system is an important and often complex issue. DrJAX provides another tool for doing this, while importantly abstracting many implementation details, such as the specific layout of data across devices.
* Based on experiments, it offers very effective parallelisation out of the box and without much tweaking.
* It appears highly extensible.

**Weaknesses:**

* DrJAX was compared with a naive Pythonic approach and a parallelizing compiler (GSPMD), but not directly compared to handwritten JAX code, where sharding and parallelisation was taken into account.
* DrJAX does not seem to have been adopted yet in any real world applications, or at least this was not mentioned if so.
* The library itself _is_ very simple, consisting of only three operations. Furthermore, it is used in this paper to perform tasks (for-loop replacement, averaging loss, tensor sharding) that do not seem exceedingly difficult to implement without the use of an additional library. The advantages of using DrJAX here could be explained further, or more complex use cases could be presented.
* The code used for experiments is not present, for instance the code that was given to the GSPMD compiler in section 4.
* Could include further discussion and comparison with alternatives (for scaling JAX applications), or if such tools and libraries exist that the authors know of.

---

### Official Review · Reviewer_cAwu · 2024-06-14
**A simple set of JAX primitives to specify distributed computation, definitely worthy of discussion at the workshop.**

**Confidence:** 4

**Summary:**

Although there are many ways to specify distributed ML problems, there is room to do better.  This paper makes a good claim to have done so.

The paper proposes a simple set of JAX primitives (broadcast, map, reduce_sum) to specify distributed ML computations.  Examples show how these primitives can indeed express several interesting computations, and it would be valuable to have this discussion at the workshop.

**Strengths:**

When reviewing this paper, the fundamental question I want to answer is "Will the workshop attendees want to use this system?"  and to get a picture of the answer, I ask "Will I use this?  Will my colleagues use this?".  I believe the answer is that we would most likely give it a go, which is certainly enough to recommend acceptance.

At the same time, there are some design decisions which trouble me.   It is quite clear that any distributed system ultimately needs to know about the nodes on which the computation is to be run.   Let us call such a description of the nodes a "mesh", roughly following existing nomenclature.

The first code snippet
```
def broadcast_double_and_sum(x):
  y = drjax.broadcast(x)
  z = drjax.map_fn(lambda a: 2*a, y)
  return drjax.reduce_sum(z)
```
Immediately has a non-functional vibe that is in stark contrast to the functional ethos of JAX.  What is "drjax"?  Presumably a module name, so whatever the description of the mesh is, it is not explicit in this listing.  The reader of this code immediately wonders where the global variables are hidden.  This does not make it easier to understand the value of the offering, it makes it harder.  I literally cannot read this code and determine what value it computes.

OK, so we quickly move to a version where the mesh is specified, and we see that only the size is really important - great, a nice property, but why hide this important information in a parse-time argument to a decorator?
```
@drjax.program(partition_size=3)
def broadcast_double_and_sum(x):
  y = drjax.broadcast(x)
  z = drjax.map_fn(lambda a: 2*a, y)
  return drjax.reduce_sum(z)
```
I believe I would much prefer to use the package in this way:
```
def broadcast_double_and_sum(mesh, x):
  y = drjax.broadcast(mesh, x)
  z = drjax.map_fn(mesh, lambda a: 2*a, y)
  return drjax.reduce_sum(mesh, z)
```
Now I can clearly see that the return value will depend on the "mesh" object; if I look inside, I may notice that its value depends only on `mesh.partition_size`; so I can reason locally about the code's behaviour.

Yes, this means that the mesh must be plumbed down through the program; but this is always the tradeoff in pure functional programming - either store a value in a global variable and lose composability, or explicitly plumb it through, just like JAX's random number keys.
```
def broadcast_double_and_sum(djmesh, x):
  y = djmesh.broadcast(x)
  z = djmesh.map_fn(lambda a: 2*a, y)
  return djmesh.reduce_sum(z)
```

Now, I get it: you're telling me what you did, and I'm saying "oh I would have done it differently".  Of course that's not a reason to reject the paper, but I am explaining why the paper seems to me to have deficiencies (alluded to in future work, because extension to "hierarchically partitioned data" is very likely going to require a redesign).

**Weaknesses:**

The paper should be much more straightforward about what currently exists.

Saying "These software frameworks generally focus on enabling parallelism for their most common use case: computation of a function’s derivative across a batch of inputs." or "an algorithm author who wishes to program over partitioned data in a parallel manner finds themselves in an awkward position" is disingenuous - yes, those easy cases are made easy in most packages, but the implication is somehow that the harder cases are not possible. This, as the paper makes clear later is not true.

Competing techniques (e.g. `jax.shard_map`) are dismissed without direct and fair comparison.  Instead of saying "Underlying ML frameworks often offer powerful parallelism primitives (e.g. `jax.shard_map`), but typically target the model developer rather than the algorithm developer"
This is not a valid distinction - DrJax solves very similar problems to `shard_map`, with a different interface - so the above sentence meaninglessly suggests that `shard_map` is not a valid comparator because of a orthogonal distinction between "mode developer" and "algorithm developer".  Further, the paper suggests that shard_map is deficient in "not abstracting away potentially nested groups of compute nodes powering computations of mapping functions."

In contrast, this paper leaves it to future work to handle "hierarchically partitioned data".   The reason this is relevant is that unless DrJax can handle all the cases that shard_map can, it is not suprising that its interface is cleaner.

**Limitations:**

Discussed above

**Suggestions:**

Some discussed above.  The primary suggestion would be to remove listing 1, and go straight to listing 2.  Of course, I would prefer you to explicitly pass all the information, but I understand that some people prefer global variables (maybe hidden in decorators or context managers) to pure functional programming.

107r: "elements of the same space" is meaningless - do you mean, have the same shape and dtype?

211l: when you say "structures", it might help readers to say that you mean the same thing as a JAX pytree, if that is what you mean, and if that's not what you mean, to say how you intend "structure" to be different from a pytree.

---

### Decision · Program_Chairs · 2024-06-18

**Decision:**

Accept (Poster)

**Comment:**

We thank the authors for their time and contribution to WANT and we are pleased to share that after the reviewing process the paper has been accepted. Congratulations! We encourage the authors to consider reviewers' feedback for the improvement of the camera-ready version. We hope to see you in person at the workshop and brainstorm on efficient training research together!